# A simple biophysical model emulates budding yeast chromosome condensation

**Tammy MK Cheng[1]\*, Sebastian Heeger[2], Raphaël AG Chaleil[1], Nik Matthews[3], Aengus Stewart[4], Jon Wright[5], Carmay Lim[6], Paul A Bates[1]\*, Frank Uhlmann[2]\***

[1]Biomolecular Modelling Laboratory, Lincoln's Inn Fields Laboratory, The Francis Crick Institute, London, United Kingdom; [2]Chromosome Segregation Laboratory, Lincoln's Inn Fields Laboratory, The Francis Crick Institute, London, United Kingdom; [3]Advanced Sequencing Facility, Lincoln's Inn Fields Laboratory, The Francis Crick Institute, London, United Kingdom; [4]Bioinformatics and Biostatistics Service, Lincoln's Inn Fields Laboratory, The Francis Crick Institute, London, United Kingdom; [5]Genomics Research Center, Academia Sinica, Taipei, Taiwan; [6]Institute of Biomedical Sciences, Academia Sinica, Taipei, Taiwan

**Abstract** Mitotic chromosomes were one of the first cell biological structures to be described, yet their molecular architecture remains poorly understood. We have devised a simple biophysical model of a 300 kb-long nucleosome chain, the size of a budding yeast chromosome, constrained by interactions between binding sites of the chromosomal condensin complex, a key component of interphase and mitotic chromosomes. Comparisons of computational and experimental (4C) interaction maps, and other biophysical features, allow us to predict a mode of condensin action. Stochastic condensin-mediated pairwise interactions along the nucleosome chain generate native-like chromosome features and recapitulate chromosome compaction and individualization during mitotic condensation. Higher order interactions between condensin binding sites explain the data less well. Our results suggest that basic assumptions about chromatin behavior go a long way to explain chromosome architecture and are able to generate a molecular model of what the inside of a chromosome is likely to look like.

\*For correspondence: tammy.
cheng@crick.ac.uk (TMKC); paul.
bates@crick.ac.uk (PAB); frank.
uhlmann@crick.ac.uk (FU)

**Competing interests:** The authors declare that no competing interests exist.

**Reviewing editor**: Taekjip Ha, University of Illinois, Urbana-Champaign, United States

## Introduction

One of the most recognizable phenomena of dividing cells is the compaction of chromatin that occurs when cells enter mitosis. In mitosis, centimeters of DNA are compacted into micrometer-sized, rod-shaped chromosomes. This allows genetic material to be packed small enough to be faithfully segregated to opposite cell halves, and compact enough to withstand the forces generated during this process. Over the past two decades, accumulating lines of evidence have indicated that the chromosomal condensin complex is a principal mediator of chromosome condensation. Condensin promotes interactions between its chromosomal binding sites (*Haeusler et al., 2008*), its depletion or genetic mutation in organisms from yeast to vertebrates leads to defective chromosome condensation, reduced mechanical chromosome stability, and consequent chromosome segregation errors (*Hirano and Mitchison, 1994*; *Saka et al., 1994*; *Strunnikov et al., 1995*; *Hagstrom et al., 2002*; *Hudson et al., 2003*; *Hirota et al., 2004*; *Oliveira et al., 2005*; *Thadani et al., 2012*). Condensin is a member of the structural maintenance of chromosomes (SMC) family of large ring-shaped multisubunit protein complexes. These are thought to bind to DNA by topological embrace (*Nasmyth and Haering, 2005*; *Cuylen et al., 2011*; *Murayama and Uhlmann, 2014*).

How condensin promotes chromosome condensation has remained unclear. Two main ideas about a possible mechanism of condensin function have been put forward. In the more traditional view,

**eLife digest** The genetic material of living things is made up of long strands of DNA. Human cells contain about two meters of DNA split between 46 chromosomes. These chromosomes carry all the instructions to build a human body. To fit all of this information inside each human cell, the DNA is wrapped around hundreds of thousands of proteins such that the chromosomes each resemble a string of beads.

Most of the time the chromosomes in a cell are only loosely arranged. But, when a cell prepares to divide into two new cells, its chromosomes become more compacted. This allows the DNA to withstand the physical forces involved when the copies of the chromosomes are pulled into the two daughter cells, and it makes it easier for the cell to handle its genetic material. If a chromosome breaks during cell division, it can result in diseases such as cancer.

Several proteins—collectively called condensins—work to compact (or condense) the chromosomes. These proteins are found in a wide range of species, but it remains poorly understood how they cause chromosomes to become more compact. Due to the technical limitations of current imaging methods, it has not been possible to directly visualize the path of the DNA strand within a compacted chromosome. However, Cheng et al. have now overcome this limitation by combining experimental analyses and computational simulations.

Cheng et al. used computer modeling to simulate a piece of chromosome that was about the same size as a chromosome from a single-celled microorganism called budding yeast. This model could accurately recreate the behavior of chromosomes as observed in non-dividing cells—and revealed that these chromosomes are in a relaxed state.

Cheng et al. then modeled what happens when condensins are introduced. As expected, the chromosomes became more compacted and the model's behavior was then validated using further experiments. This predicted that condensin complexes, bound to regions along the chromosome's length, interact to form pairs that continually separate and form new pairs with other condensins; and that these 'dynamic pairwise' interactions compact the chromosome. The current model describes a relatively small chromosome and, in the future, extending the model to larger chromosomes could shed insight.

condensin forms higher order assemblies within chromosomes, thought of as part of a chromosome scaffold, to which loops of DNA are attached. This view is supported by cytological observations of condensin localization and early biochemical analyses, but also by recent simulations of chromatin interactions within human chromosomes (*Maeshima and Laemmli, 2003*; *Swedlow and Hirano, 2003*; *Naumova et al., 2013*; *Maeshima et al., 2014*). In a contrasting model, condensin has been proposed to act by providing DNA interactions between its chromosomal binding sites in a more stochastic manner, without the need to engage into higher order assemblies. This idea is supported by measurements of the biophysical properties of chromosomes and high resolution electron tomographic imaging of chromosomes in their close to native state (*Poirier and Marko, 2002*; *König et al., 2007*; *Thadani et al., 2012*; *Maeshima et al., 2014*). However, technical limitations mean that it remains a hitherto insurmountable challenge to directly visualize the path that a DNA strand takes inside a chromosome and how and where condensin acts.

In this study, we use an *ab initio* coarse-grained Brownian dynamics simulation of a budding yeast chromosome to explore chromatin behavior during chromosome condensation. We make no assumptions about chromatin behavior other than known physical properties of a nucleosome chain (*Robert, 1995*; *Grassia and Hinch, 1996*; *Luger et al., 1997*). We then introduce condensin to provide intrachromosomal interactions. These interactions are modeled to be (i) stochastic pairwise interactions between two chromosomal binding sites (Type I model), or (ii) stochastic but able to engage in higher order assemblies where more than two condensin binding sites meet (Type II model). We compare the predictions from these simulations with experimental chromatin proximity data obtained by 4C analysis on budding yeast chromosome 5 and with other measured biophysical chromosome properties. This analysis shows that stochastic pairwise interactions of a chromatin chain, mediated by condensin, provide a close fit to observed chromosome behavior in budding yeast.

## Results

### A physics-based computational model of a budding yeast chromosome

We constructed a computational model to simulate emergent behavior of a large chromosome fragment, consisting of a string of 2000 nucleosomes, representing approximately 300 kb in genomic distance. This is longer than the two smallest budding yeast chromosomes, and similar for example, to the length of the long arm of budding yeast chromosome 5. We applied a 'bead-spring' model in which the chromatin chain is represented as a series of nucleosome beads, joined by DNA linkers whose dynamic behavior is approximated as springs (*Figure 1A*). The nucleosome string thus behaves as a chain, where the DNA linkers regulate the movement of joined nucleosomes according to Hooke's law. The movement of each nucleosome bead follows a Brownian dynamic trajectory, approximating the solution state in the nucleus. Nucleosomes exclude each other in space though, during our simulations, the linker DNA can cross itself with a small probability, which, under in vivo conditions would be achieved by the enzyme topoisomerase II. A weak force corrects the angle at which DNA emerges from the nucleosome surface (*Luger et al., 1997*; *Bednar et al., 1998*; *Engelhardt, 2007*) (see 'Materials and methods' for details of the model).

To analyze the impact of a chromatin crosslinker on chromosome behavior, we introduced condensin binding sites along the chromatin chain. This was guided by the experimentally visualized condensin distribution along budding yeast chromosomes using chromatin immunoprecipitation (*Wang et al., 2005*; *D'Ambrosio et al., 2008*), which revealed condensin-enriched peaks with an average distance of approximately 10 kb (*Figure 1B*). We therefore assigned a condensin binding site approximately every 10 kb along the chromatin chain. *Figure 1C* shows a relaxed conformation of the resultant chromatin chain at the beginning of our simulations. If, on their stochastic trajectories, two condensin binding sites come within 40 nm of each other, an interaction is established between them. While we are as yet naïve about how condensin bridges two binding sites, this assumption lies at the heart of the majority of models for condensin action. The attraction radius of 40 nm is based on condensin's molecular architecture (*Anderson et al., 2002*), the ability of chromosomes to compact was insensitive to its exact value. Once established, the interactions dissolve with a model-specific dissociation probability, equivalent to an off-rate of the dynamically chromosome-bound condensin complex (*Gerlich et al., 2006*).

We compared two distinct modes of how condensin might act. In our Type I model, each condensin binding site can interact with exactly one other site at a time, thus leading to stochastic pairwise interactions between chromatin segments (*Figure 1D*). In the Type II model, each condensin binding site can interact with up to two others, thereby allowing the formation of higher order condensin binding site assemblies. This model provides a means to interrogate chromosome behavior based on first principles.

### Native-like chromosome dimensions, tunable by the condensin dissociation rate

We first compared chromosome dimensions in our model to those observed in vivo. To do so, we used a budding yeast strain in which two loci on the chromosome 5 long arm, at 144 kb distance from each other, were fluorescently marked in distinct colors using the TetO/TetR-YFP and LacO/LacI-CFP systems, respectively (*Rohner et al., 2008*). This allowed us to measure their in vivo 3D distance with great precision, yielding interphase distances consistent with previous measurements using loci marked with the same fluorophore (*Guacci et al., 1994*; *Bystricky et al., 2004*; *D'Ambrosio et al., 2008*). In G1-arrested cells, the mean distance between the loci was 670 nm (*Figure 2A*), which was similar to the distance of similarly spaced loci in the relaxed starting configuration of our model (*Figure 2B*). This striking correspondence suggests that interphase chromatin in vivo adopts a configuration of similar dimensions as compared to the dimensions of an unconstrained nucleosome fiber.

To see how condensin action impinges on chromosome behavior in our computational model, we allowed either Type I or Type II interactions between condensin binding sites and followed the marker distance over time. At a high dissociation probability ($10^{-3}$, i.e., a probability of $10^{-3}$ per simulation step that an existing interaction is lost), chromosome dimensions remained largely unchanged. In case of Type I interactions, the marker distance fluctuated around 600–700 nm (*Figure 2C*). Allowing Type II interactions resulted in a slightly more compacted chromosome and marker distances between

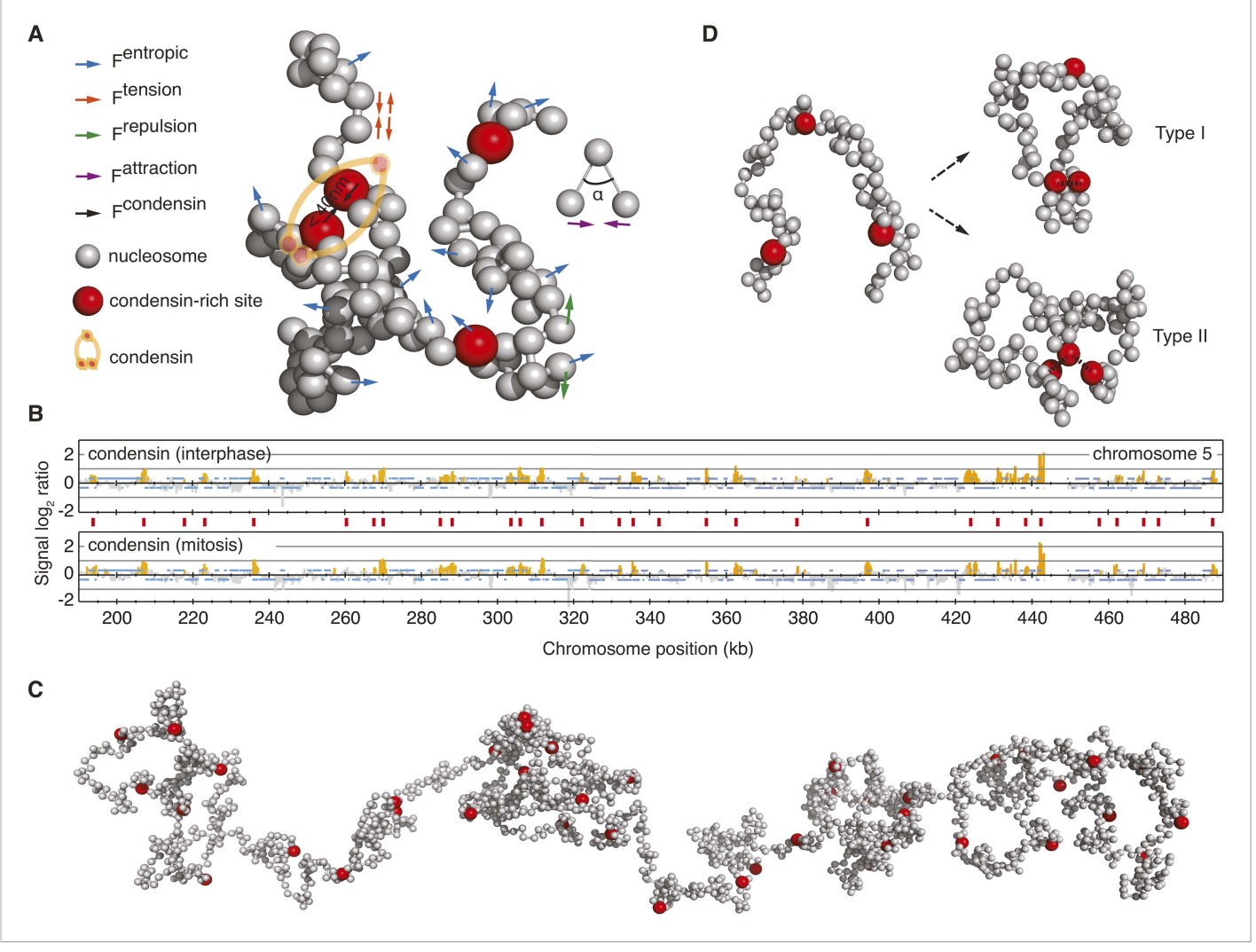

**Figure 1**. A computational chromosome model. (**A**) Schematic of the forces enacted during simulation. Inter-joined grey beads represent nucleosomes, condensin binding sites are highlighted in red. F$^{entropic}$ (blue arrows) move each bead in a Brownian dynamic trajectory, constrained by F$^{tension}$ (red arrows), a spring force that connects nucleosome beads, F$^{repulsion}$ (green arrows) that avoids overlaps between beads, F$^{attraction}$ (purple arrows), a weak force that corrects the angle at which DNA linkers emanate from the nucleosomes and F$^{condensin}$ that maintains the vicinity of two condensin binding sites, if they meet. (**B**) Condensin localization along a 300 kb region on the right arm of budding yeast chromosome 5, showing condensin binding sites (red vertical lines) at approximately 10 kb intervals. (**C**) View of a relaxed starting conformation of the simulated 300 kb nucleosome chain. (**D**) Illustration of Type I and Type II interactions, where pairs of condensin binding sites interact, or where one binding site interacts with up to two others, respectively.

The following figure supplements are available for figure 1:

**Figure supplement 1**. Nucleosome displacement over time in our computational chromosome model.

**Figure supplement 2**. α angle distribution of DNA entry and exit from nucleosomes in simulated chromosomes.

500—600 nm. Taken together, chromatin interactions with a high off-rate are compatible with interphase chromosome dimensions.

In mitotically arrested cells, the mean in vivo distance between the two loci decreased to 497 nm, corresponding to an approximately 25% length compaction, equivalent to a just over twofold volume compaction, which depended on the condensin complex (*Figure 2A*). The compaction ratio was slightly less than reported in previous studies (*Guacci et al., 1994*; *Strunnikov et al., 1995*; *Vas et al., 2007*; *D'Ambrosio et al., 2008*). A likely reason is our use of two colors to distinguish the two loci.

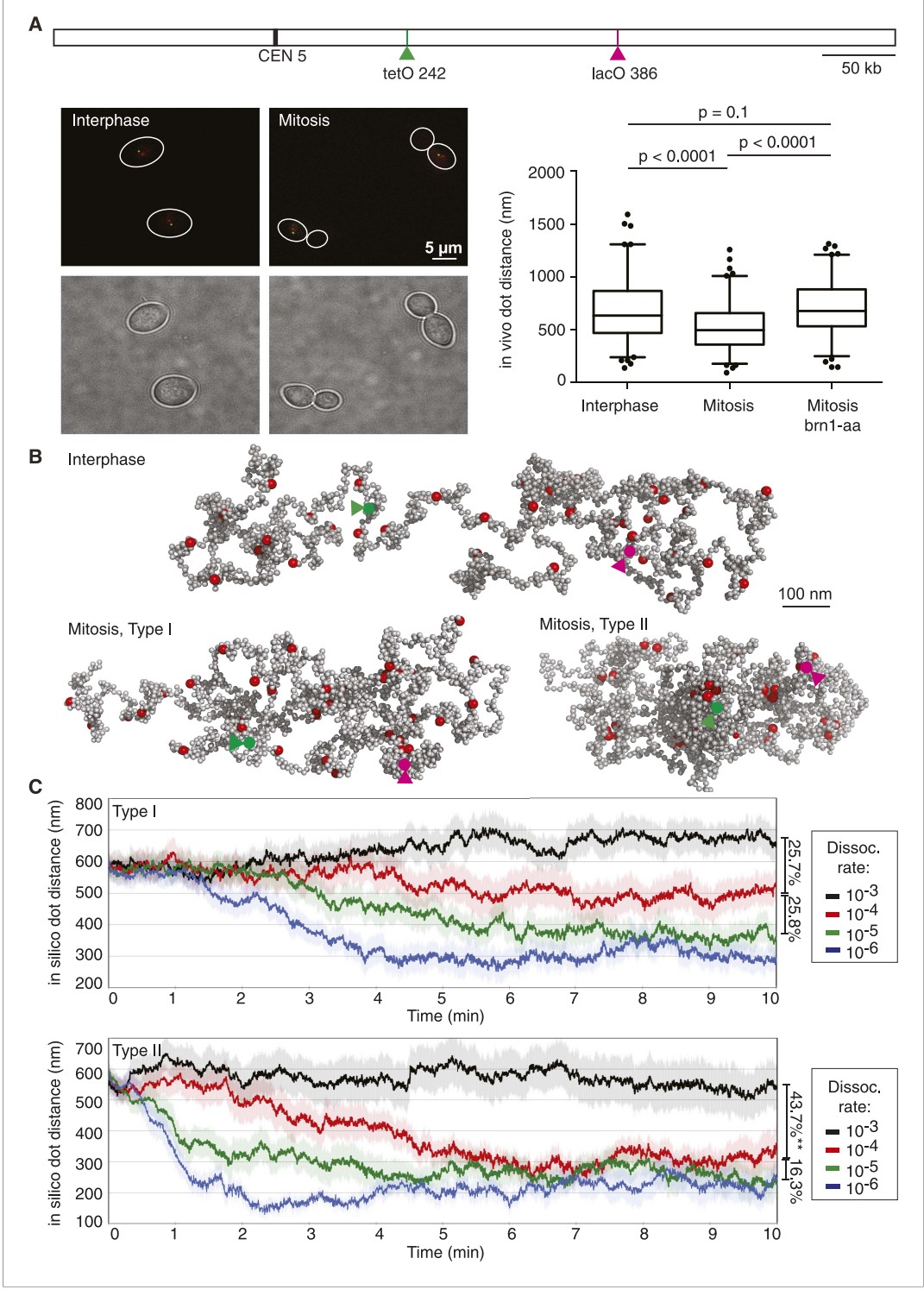

Figure 2. Chromosome dimensions during experimental and computational condensation. (**A**) Scheme showing the location of the two loci whose distance was recorded in vivo and during each simulation. Example micrographs of wild type cells in interphase and mitosis are shown, together with a graph depicting the median, upper, and lower quartiles, with whiskers at 2.5 and 97.5%, outliers also plotted, for both wild type strains in interphase and mitosis, as well as for a strain in mitosis in which condensin has been depleted from the nucleus using the *brn1-aa* allele (*Haruki et al., 2008*; *O'Reilly et al., 2012*; *Charbin et al., 2014*). Statistical significance of the differences was assessed using a Wilcoxon–Mann–Whitney test. (**B**) Example of an interphase conformation (Type I model, condensin

*Figure 2. Continued*

interaction dissociation rate $10^{-3}$) of a simulated chromosome, the two marker loci are highlighted, as well as mitotic conformations (dissociation rate $10^{-4}$) generated by the Type I and Type II models. (**C**) Traces of marker distance over time after the dissociation rates were set to the indicated values at t = 0. Shown are the mean and the standard error of 30 simulations. The linear compaction ratios are noted for the indicated comparisons.

The following figure supplement is available for figure 2:

**Figure supplement 1**. Traces of marker distances over time in the Type II model at dissociation probability $5 \times 10^{-4}$.

---

This allows us to assign a discrete distance even to close foci in mitosis that would have been considered to be at zero distance due to the resolution limit in single color observations. We note that a twofold volume compaction during budding yeast chromosome condensation is similar to what is observed for example, when comparing human interphase and mitotic chromosomes (*Mora-Bermúdez et al., 2006*). When we reduced condensin's dissociation probability in our simulations, akin to the stabilization of condensin binding to chromosomes that has been observed as human cells enter mitosis (*Gerlich et al., 2006*), chromosomes began to compact. A 10-fold reduction of dissociation probability in the Type I model resulted in a 25% length compaction, comparable to what we observed in vivo (*Figure 2B,C* and *Video 1*). Further reduction of the dissociation probability led to further gradual compaction. In the Type II model, a 10-fold reduction of the dissociation probability caused a length compaction of over 40%, more than what is observed in vivo (*Figure 2B,C* and *Video 2*). A smaller reduction of the dissociation probability by only twofold, to $5 \times 10^{-4}$, led to a bistable behavior of the chromosome, its compaction fluctuating between open and closed equilibrium states of similar dimensions as those obtained at $1 \times 10^{-4}$ and $1 \times 10^{-3}$, respectively (*Figure 2—figure supplement 1*). These observations suggest that the half-life of condensin-mediated interactions along a chromatin fiber has the potential to determine the chromosome condensation status. Type I interactions allow compaction to be tuned within a physiological range. In contrast, Type II interactions result in a more stepwise compaction, the degree of which exceeds what we observed in vivo.

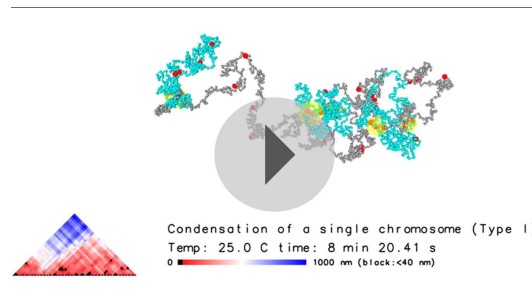

**Video 1.** Condensation of a single chromatin chain in the Type I model. The video shows two representative stages of simulated chromosome condensation: (1) the initial extended chromosomal structure (5 s are shown), followed by the 8th min, when the chromosome has reached a compacted steady-state (20 s are shown). Nucleosomes are shown as grey spheres and condensin binding sites are in red. If more than two condensin binding sites come within 40 nm of each other, they are highlighted by a yellow sphere to indicate 'rosette' formation. Chromatin loops that connect condensin binding sites within rosettes are tinted cyan. A balanced co-existence of rosette and web-like structures in the compacted mitotic stage becomes apparent.

## Computational and experimental intrachromosomal interaction maps

To further evaluate our chromosome model, we compared simulated with experimental intrachromosomal contact frequency maps. We utilized the 4C (circularized chromosome conformation capture) (*Dekker et al., 2013*) technique combined with high throughput sequencing to generate high resolution interaction maps of 4 loci along the budding yeast chromosome 5 long arm (*Figure 3A*). This revealed an interaction pattern in interphase that was largely contained within approximately 100 kb around the view point (*Figure 3B*). Two condensin binding sites showed increased local interactions, compared to two viewpoints that were relatively depleted of condensin. Intrachromosomal interactions markedly increased in mitotic cells, both in the vicinity of the view point as well as longer range interactions beyond 100 kb. This increase depended on the condensin complex and was largely reduced when condensin was depleted

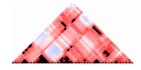

**Video 2.** Condensation of a single chromatin chain in the Type II model. As *Video 1*, but the simulation followed the Type II model. Now the rosette-like topologies become dominant and the overall structure is densely packed.

from nuclei using the *brn1-aa* allele (*Charbin et al., 2014*). Together this suggests that condensin binding sites are hubs of intrachromosomal interactions, and that these are augmented in mitosis.

If condensin promotes interactions between its binding sites, these might become detectable as interaction peaks in our 4C interaction maps. However, such peaks were not clearly discernible (*Figure 3B*). A possible explanation for this is that condensin-enriched sites are relatively broad features and their approximately 10 kb spacing is close to the resolution limit of the 4C technique, given by the sizes of the HindIII restriction fragments used in our analysis that are in a similar size range. Alternatively, we cannot exclude that condensin engages in interactions not only between its binding sites, but also with chromatin features between binding sites, for example, histones (*Tada et al., 2011*). For ease of analysis, in our present study, the computational analysis focuses on interactions between condensin binding sites.

The resultant simulated intrachromosomal contact frequency map derived from our Type I chromatin interactions showed striking qualitative resemblance to the experimental interaction maps. The similarities extended to (i) the distribution of interactions and their reach to approximately 100 kb, (ii) a greater number of interactions that emanate from condensin binding sites, (iii) an increase of interactions in mitosis (*Figure 3C*). The Type II model produced a contact frequency distribution of a broader shape, extending farther from the view point than observed, especially in mitosis. As a quantitative measure to compare the experimental and computational interaction maps, we recorded the percentage of intrachromosomal interactions that extend beyond 100 kb. While overall intrachromosomal contacts increased in mitosis, the fraction of interactions beyond 100 kb was in a similar range between interphase and mitosis (*Figure 3D*). The same was observed in simulations using the Type I model, while the Type II model predicts the appearance of significantly more mitosis-specific long-range interactions than observed.

## Dynamic web vs stable rosette characteristics of chromatin

We next studied the implications of the Type I and Type II models on the pattern and dynamics of intrachromosomal interactions. Pairwise interactions between condensin binding sites in the Type I model gave rise to a loose web-like architecture spanning much of the chromosome volume in interphase (*Figure 4A*). Interactions within the web were dynamic and frequently interchanging. In mitosis, rosette-shaped foci formed that contained more than two binding sites within condensin's interaction radius of 40 nm. These foci were maintained for short periods of time by alternating pairwise binding site interactions, before they dissolved again (*Figure 4B* and *Video 1*).

An interphase Type II chromosome was only in part characterized by a web-like architecture. Instead, over one third of its length was organized in rosettes in which more than 2 condensin binding sites interacted (*Figure 4A*). In mitosis, most of the chromosome consisted of extended rosette structures that persisted for longer and involved a greater number of condensin binding sites as compared to the Type I model (*Figure 4B* and *Video 2*). Many chromosomal activities, for example, gene regulatory interactions or recombination events, are thought to involve rapid genome scanning for correct contacts which we expect is facilitated by a dynamic chromosome organization.

## Simulated and experimental polymer characteristics

In addition to the mean distance between marker loci, used above to benchmark chromosome dimensions, we also compared the distance distributions between cells in a population to those from the computer simulations. The kurtosis (K) is a dimensionless quantity that describes the difference of a distribution from normal, and is a distinctive feature of polymer models (*Gennes, 1979*; *Balanda and MacGillivray, 1988*; *Barbieri et al., 2012*). *Figure 5A* plots K values for a range of chromosomal distances, observed at 100 timepoints in 30 repeats of our chromosome simulations. They range

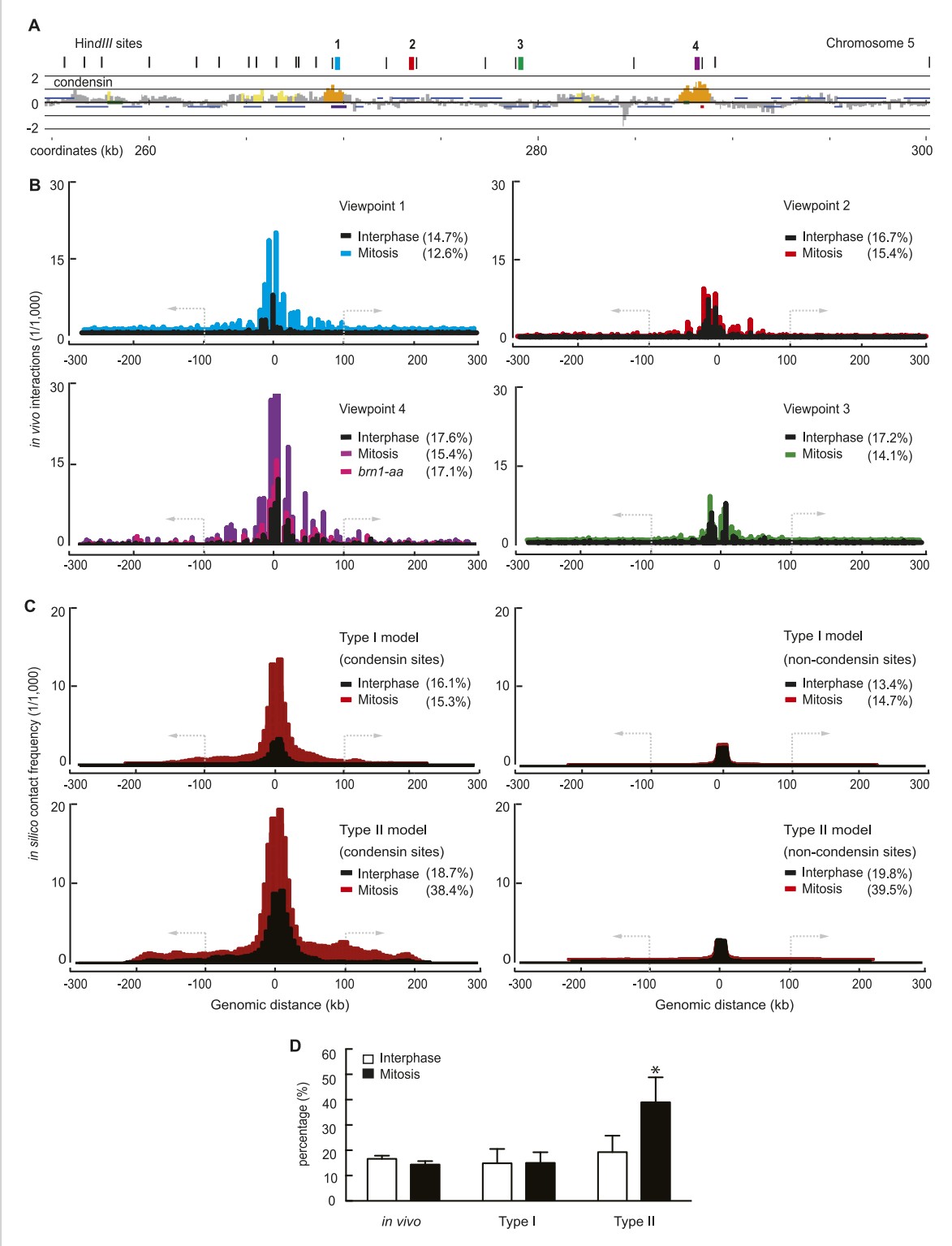

**Figure 3**. Experimental and computational intrachromosomal interaction frequency maps. (**A**) Close-up of the chromosomal viewpoints selected for 4C analysis. Condensin localization along part of the chromosome 5 right arm is shown together with genomic HindIII recognition sites and the four 4C view points that do (1 and 4) or do not (2 and 3) contain a condensin binding site. (**B**) Experimental 4C interaction maps of the four regions, in both interphase and mitosis. Shown is also a 4C map of region 4 in mitosis after condensin has been depleted from the nucleus using the *brn1-aa* allele. The y-axis shows sequencing read counts normalized to the total number of mapped reads in each sample. The percentage of interactions that extend farther than 100 kb from the viewpoint is indicated. (**C**) Averaged computational intrachromosomal interaction maps of 6 viewpoints within 50 kb from the chromosome ends,
*Figure 3. continued on next page*

*Figure 3. Continued*

on or between condensin binding sites, generated using both the Type I and Type II model and sampled over 1000 time points and 30 simulations in interphase and mitosis (condensin interaction dissociation rates $10^{-3}$ and $10^{-4}$, respectively). The y-axis shows interaction frequencies of the viewpoints normalized to all interactions. (**D**) Percentage of interactions that extend beyond 100 kb from the viewpoint under the indicated conditions. The mean of the four experimental fragments, or of the simulated distributions, is shown together with the standard deviation. *p < 0.0001, Wilcoxon–Mann–Whitney test.

between 2.2 and 2.8, with the Type II model showing slightly smaller values compared to Type I, indicative of distance distributions that have a broader peak than a normal distribution (K = 3). We compared this to the K of the experimental distance distribution (*Figure 2A*) and to a published dataset of in vivo distance measurements in budding yeast (*Bystricky et al., 2004*). The observed K values agreed well with those seen in our simulations, they lay somewhat closer to the values seen in the Type I as compared to the Type II model. Thus, the K value of the simulated distance distributions agrees with those observed in vivo.

A defining characteristic of polymer models is the bending rigidity (or persistence length, $L_p$). We estimated $L_p$ of the simulated structures from both Type I and II models using an orientational correlation function (see 'Materials and methods'). In both models, $L_p$ increases with greater genomic distances, as expected from loop polymers (*Strobl, 1997*). At smaller genomic distances, up to ~150 kb, the two models show comparable persistence lengths, with values in agreement with experimental $L_p$ estimates in budding yeast of 58–134 nm, based on physical distance and 3C interaction frequency measurements in this distance range (*Dekker, 2008*) (*Figure 5B*). At the smallest distances in our simulations the persistence length approaches that of the free chromatin chain without loops, again

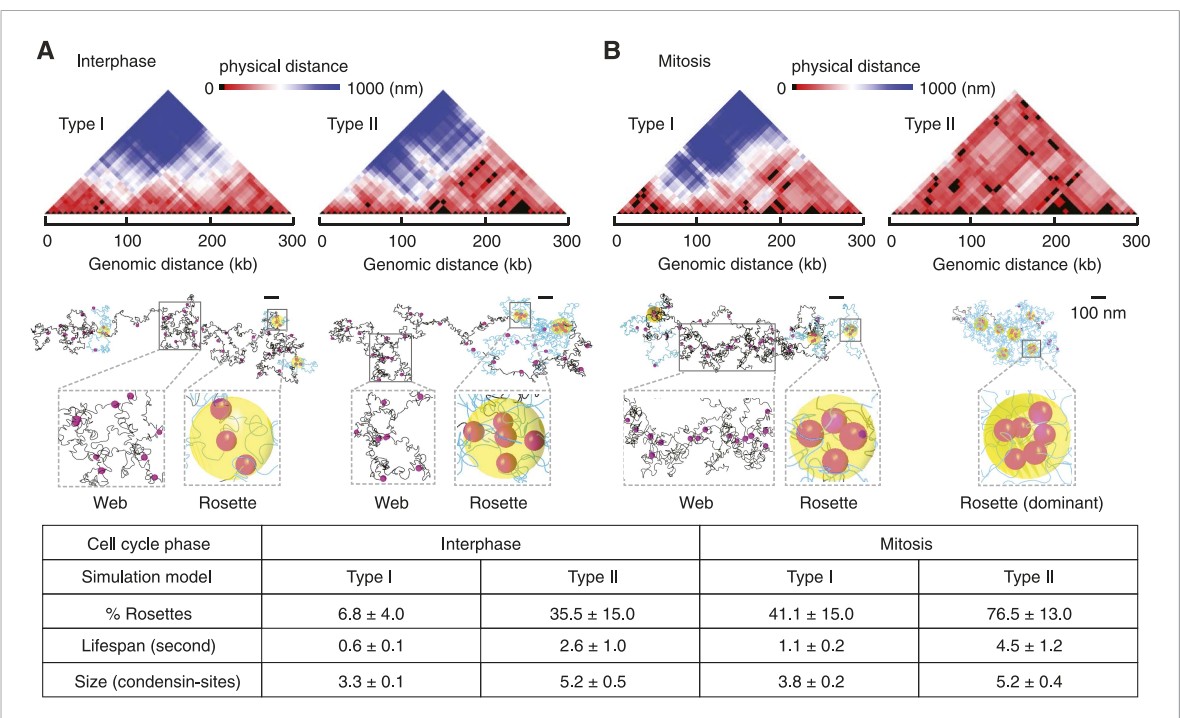

| Cell cycle phase | Interphase | | Mitosis | |
|---|---|---|---|---|
| Simulation model | Type I | Type II | Type I | Type II |
| % Rosettes | 6.8 ± 4.0 | 35.5 ± 15.0 | 41.1 ± 15.0 | 76.5 ± 13.0 |
| Lifespan (second) | 0.6 ± 0.1 | 2.6 ± 1.0 | 1.1 ± 0.2 | 4.5 ± 1.2 |
| Size (condensin-sites) | 3.3 ± 0.1 | 5.2 ± 0.5 | 3.8 ± 0.2 | 5.2 ± 0.4 |

**Figure 4**. Web and rosette characteristics of the intrachromosomal interaction pattern. (**A**) 3D distance maps of the condensin binding sites, a snapshot of an interphase simulation is shown. Each position along the x axis represents a condensin binding site, the color-coded distance between each is shown above. The corresponding snapshot of the chromosome is partitioned into web (grey) and rosette (blue) compartments. Yellow spheres highlight the core of the rosette structures where more than two condensin binding sites are in proximity. (**B**) as (**A**), but snapshots are shown from simulations in mitosis. A summary of the percentage, life-span, and size of rosette structures within the chromosome, averaged over 3000 time intervals and 30 simulations is given in the table.

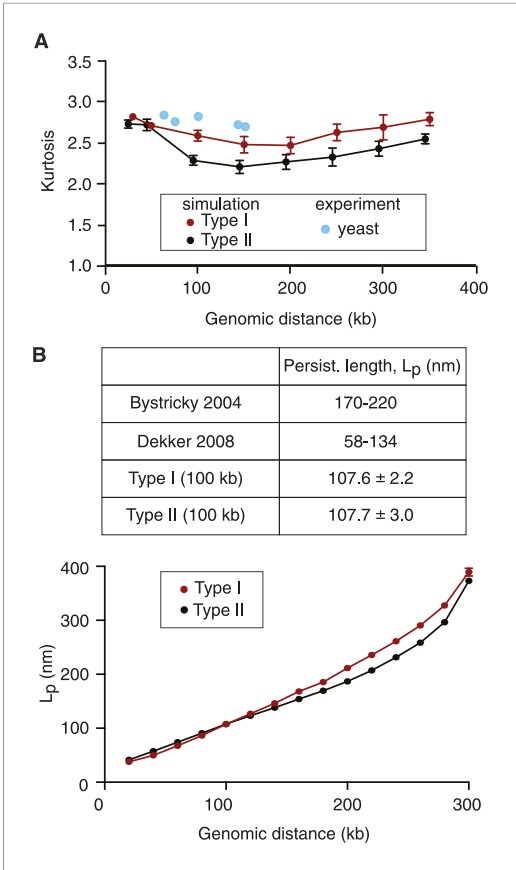

**Figure 5**. Polymer characteristics of simulated and native budding yeast chromatin. (**A**) Kurtosis values calculated from the simulations and experimental measurements in interphase. Experimental data were from *Figure 2* and from published measurements (*Bystricky et al., 2004*). (**B**) The persistence length $L_p$ of chromatin in the Type I and Type II model as a function of genomic distance. 100 chromosome conformations of each model were exhaustively sampled with the orientation correlation function, the means and standard deviations of $L_p$ are plotted. The values in the table are from the 100 kb cut-off, a range similar to that used in the experimental measurements (*Bystricky et al., 2004*; *Dekker, 2008*). DOI: 10.7554/eLife.05565.012

showing values in line with an experimental $L_p$ estimate for the local persistence length of budding yeast chromatin of approximately 30 nm (*Hajjoul et al., 2013*). At distances greater than 150 kb, the persistence lengths of Type I and II chromosomes becomes significantly distinct. To our knowledge, there is currently no experimental estimate for $L_p$ at these distances, which may be in part due to the difficulty of constraining $L_p$ values by fitting experimental data to analytical functions at these distances. We therefore do not know which model better describes the rigidity of yeast chromosomes at greater genomic distances.

## Chromosome individualization during condensation

The 16 budding yeast chromosomes lie in close contact to each other in the nucleus (*Duan et al., 2010*). If chromosome condensation indeed occurs by stochastic pairwise interactions between condensin binding sites, how does condensin discriminate intrachromosomal interactions that condense a chromosome from interchromosomal interactions that lead to unproductive chromosome crosslinks? To explore this, we simulated chromosome condensation of two 300 kb long chromatin chains lying adjacent to each other (*Figure 6* and *Videos 3, 4*). As expected, we observed condensin-mediated interactions both within and between chromosomes. In the Type I model, intrachromosomal interactions became dominant over time and the two chromosomes formed individual entities, while maintaining occasional dynamic contact between their surfaces. Starting from the same chromosome positions, the Type II model also displayed a tendency for chromosomes to individualize. However, the more stable nature of rosette-like interaction hubs that formed both within and between chromosomes often maintained the two chromosomes in an entangled state and prevented their complete separation.

These results show that condensin-mediated interactions individualize chromosomes, as condensin is more likely to encounter binding sites along the same nucleosome string compared to binding sites on two independently moving chains. Dynamic pairwise interactions between condensin binding sites are better able to prevent persistent cross-linking of chromosomes than rosette-like interaction hubs.

## Discussion

In this study, we explore chromosome architecture by building a simple computational model of a budding yeast chromosome, consisting of coarse-grained representations of the two essential elements for condensation, nucleosomes, and a chromatin cross-linking protein, modeled in our simulations to follow the distribution and behavior of condensin. We compare the emergent model behavior to a panel of existing and new in vivo measurements. Previous models have studied the behavior of short chromatin pieces, including details about nucleosome structure (*Woodcock et al., 1993*; *Grigoryev et al., 2009*; *Schlick and Perišić, 2009*; *Diesinger and Heermann, 2010*).

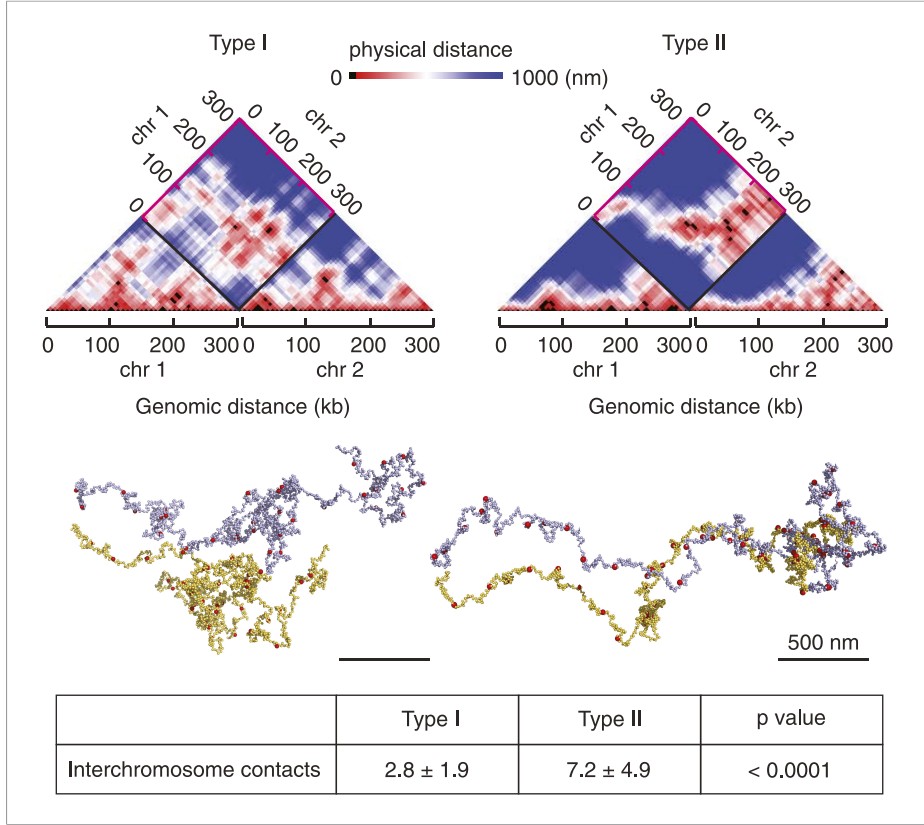

**Figure 6.** Chromosome individualization during condensation. Snapshots are shown of chromosomes and their 3D distance maps, after 5 min of simulated chromosome condensation of two adjacent chromosomes using the Type I and Type II models. The average number of interchromosome contacts over the 10 min condensation timecourse are indicated. Statistical significance of the difference was assessed using a Wilcoxon–Mann–Whitney test.

Other models were aimed at describing whole genome organization in the yeast nucleus (*Duan et al., 2010*; *Tjong et al., 2012*; *Tokuda et al., 2012*; *Wong et al., 2012*), or chromosome behavior at a larger scale but lower resolution (*Barbieri et al., 2012*; *Brackley et al., 2013*; *Naumova et al., 2013*; *Giorgetti et al., 2014*). A defining feature of our coarse-grained nucleosome polymer model is that it allows higher order chromosome structure to arise from first principles, the physics of a solvated polymer chain driven by non-specific entropic forces that generate Brownian motion. The model uses a simulated 10 nm nucleosome chain, to which we add two different scenarios of how the chromosomal condensin complex might act. The emergent chromosome architecture is strikingly in line with numerous features that we and others have measured in vivo, offering quasi-molecular insight into what the inside of a chromosome might look like.

A first surprise came from the realization that the dimensions of a relaxed computational nucleosome chain are similar to those determined experimentally for interphase budding yeast chromatin. The reason for this surprise is that a nucleosome chain is often portrayed as beads on a straight line. However, a straight line presents a highly ordered state and entropic forces work to fold this into a much more irregular configuration. The angle at which DNA emanates from a nucleosome further promotes generation of a more rugged path of the nucleosome chain. These results suggest that interphase chromatin exists in a relatively relaxed state inside a budding yeast nucleus and that there may be no need to invoke major forces, scaffolds or other organizing principles to explain its packing. The modeled chromatin packing density, when scaled up to the entirety of the budding yeast genome, allows its comfortable fit within the budding yeast nucleus (see 'Materials and methods').

We explored the consequences of condensin-mediated interactions along the chromatin chain. Our 4C results suggest that condensin binding sites are hubs of intrachromosomal interactions, and

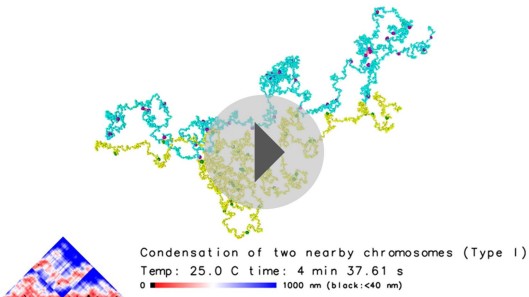

**Video 3.** Condensation of two nearby chromatin chains in the Type I model. 25 seconds of two chromatin chains compacting next to each other using the Type I model are shown, illustrating chain separation during condensation (chain 1, nucleosomes in light blue, condensin binding sties in red; chain 2, nucleosomes in yellow, condensin binding sites in green).

that these interactions are promoted by the condensin complex. We do not yet know how condensin promotes intrachromosomal interactions. One condensin ring might sequentially topologically entrap DNA at two of its binding sites, or two condensin complexes at their respective binding sites might interact with each other. In addition, condensin might engage in interactions with chromatin or nucleosomes at places distinct from its primary binding sites, and such additional contacts might display similar or different characteristics from topological DNA interactions. While the molecular biology underlying condensin's binding mechanism is important to explore, our simulations are oblivious to the molecular details that underlie the interactions. Instead, we found that the half-life and topology of interactions that each condensin binding site can engage in profoundly affects chromosome behavior.

In our Type I model, interactions are pairwise, while the Type II model allows each condensin binding site to interact with up to two others. The latter mode allows the seeding and propagation of higher order assemblies of condensin binding sites, in a way that is often portrayed in models of condensin action. Having compared the two models in our simulations, we conclude that Type I interactions fare better at generating a wide spectrum of chromosomal features that match in vivo observations, including genomic to physical distance distributions and intrachromosomal interaction maps in both interphase and mitosis. The Type I interactions also perform better in individualizing neighboring chromosomes during their mitotic condensation. It has been suggested, on theoretical grounds, that a 'weak chromatin glue' is required to allow chromosome individualization during condensation (*Marko and Siggia, 1997*). In our Type I model, this weakness is achieved by ongoing dynamic reorganization of the chromatin web, compared to the more stable rosette structures formed by Type II interactions. From a physical point of view, a dynamic web structure allows the small advantage of intrachromosomal interactions, arising from the physical continuity of the chromatin chain, to separate chromosomes over time. In other words, it provides a safeguard to reduce the degree of entanglement, should it occur. We cannot exclude that both Type I and II interaction modes occur simultaneously on chromosomes. Indeed, rosette-like interaction hubs transiently form during simulations based on Type I interactions. It remains possible that modifications to condensin, for example due to different levels of phosphorylation, could alter the balance between the Type I and II mechanisms. In any event, our simulations suggest that maintaining a dynamic aspect of chromatin interactions confers advantages during the chromosome condensation process.

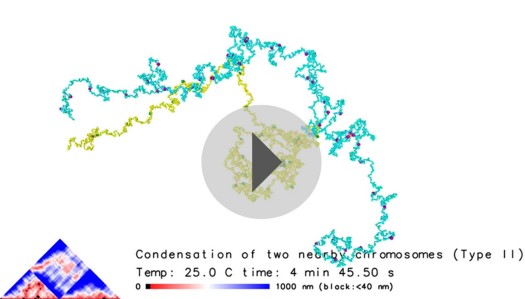

**Video 4.** Condensation of two nearby chromatin chains in the Type II model. As *Video 3*, but using the Type II model. The two chromatin chains fail to separate during condensation. The full-length, high resolution versions of all the videos can be found with the digital object identifier doi:10.5061/dryad.78622 at http://datadryad.org.

Is chromosome architecture in budding yeast, and potentially other organisms, sufficiently described by a self organizing chromatin chain, constrained by condensin-mediated interactions? On the one hand, a web-like chromosome architecture is consistent with biophysical and microscopic analyses also of higher eukaryotic chromosomes (*Poirier and Marko, 2002*; *König et al., 2007*; *Nishino et al., 2012*; *Thadani et al., 2012*). On the other hand, our model is doubtless an oversimplification. Condensin is only one of at least three SMC family complexes in eukaryotes, all of which are likely to act as

chromatin crosslinkers by promoting DNA interactions. In addition, many organisms contain more than one type of condensin complex, whose chromosomal distributions appear to be distinct. Condensin might also interact with histones and histones in turn engage with each other, while both types of contact are regulated by posttranslational modifications. Linker histones, in organisms that encode them, alter the flexibility of the nucleosome chain. We expect that a plethora of inputs have the potential to modulate or add parameters in our model. What our study shows is that simple assumptions based on first principles go a long way towards explaining chromosome behavior. It will be fascinating to extend similar simulations to much longer nucleosome chains. Will topologically associating domains, that are seen in large chromosomes (*Dixon et al., 2012*; *Mizuguchi et al., 2014*), emerge from subtle inhomogeneities in the condensin distribution? Which additional principles will have to be incorporated into the model to make an iconic metazoan X-shaped mitotic chromosome gain shape?

## Materials and methods

### Yeast cell culture and 4C protocol

The detection of intrachromosomal interactions along budding yeast chromosomes was based on previously published protocols (*Singh et al., 2009*; *Splinter et al., 2012*). Cells were grown to mid log phase and arrested in G1 (interphase) by α-factor or a-factor treatment (*O'Reilly et al., 2012*), or in metaphase by nocodazole treatment. Rapamycin treatment to deplete nuclear condensin using the anchor-away technique (*Haruki et al., 2008*) was performed as described (*Lopez-Serra et al., 2013*; *Charbin et al., 2014*). Uniform arrest was confirmed by cell morphology and FACS analysis of DNA content. 400 ml of culture were crosslinked at room temperature with 1% formaldehyde for 15 min followed by quenching with 125 mM glycine for 5 min. Cells were washed twice in ice cold TBS and resuspended in 1 ml FA buffer (50 mM HEPES/KOH pH7.9, 140 mM NaCl, 1 mM EDTA, 1% Triton X-100, 0.1% sodium deoxycholate, protease inhibitors). Cells were broken in a multi bead shocker (Yasui Kikai Corporation, Japan) using acid-washed glass beads. Chromatin was pelleted by centrifugation. Samples were taken at each step to assess the chromatin state by agarose gel electrophoresis and by quantitative real time PCR. The chromatin pellet was resuspended in 500 μl NEBuffer 2 (New England Biolabs, Ipswich, MA). SDS was added to a final concentration of 0.1% and extraction was allowed for 10 min at 65°C followed by quenching of the SDS with 1% Triton X-100. 0.1 mg/ml RNase A was added and incubated for 2 hr at room temperature. Now, 2000 units HindIII (New England Biolabs) were added and the digest kept at 37°C with frequent mixing. After 2 hr the same amount of enzyme was added again and incubation continued over night. In the morning, the enzyme was inactivated by incubation at 65°C for 20 min.

For proximity ligation, the crosslinked chromatin was diluted 20-fold (to 10 ml) with DNA ligase buffer and incubated with 1600 units of T4 DNA ligase (New England Biolabs) at 16°C for 2 hr. The crosslinks were now reversed by addition of 100 μl proteinase K (20 mg/ml) and 1% SDS and incubation at 65°C over night. DNA was purified by Phenol/Chloroform extraction and precipitated by addition of 200 mM NaCl and 70% (final) of ice cold ethanol. The resulting 3C library was dissolved in 500 μl 10 mM Tris/HCl pH 7.9.

440 μl of the above 3C library were adjusted to NEBuffer DpnII and digested with 100 units DpnII (New England Biolabs) for 4 hr at 37°C, followed by heat inactivation. Dilution, a second round of proximity ligation and DNA purification were repeated as above. The resulting final 4C library was dissolved in 500 μl 10 mM Tris pH 7.9.

1 μl of the 4C library was used as template for PCR amplification using oligonucleotide primers adjacent to pairs of HindIII and DpnII sites on the long arm of budding yeast chromosome 5. Aliquots of the PCR reactions were analyzed by agarose gel electrophoresis, the remainder was applied to QIAquick PCR Purification Kit (Qiagen, Netherlands) for DNA purification.

In preparation for sequencing, the DNA samples were end repaired, poly-A tailed and Single End Adapters (Illumina, San Diego, CA) were ligated. The manufacturers 'ChIP-Seq' protocol was adjusted for our samples. We used Agencourt AMPure XP beads (Beckman Coulter, Brea, CA) at a 0.8× ratio to remove adapter dimers after ligation and replaced the Illumina Phusion enzyme with the Kapa HiFi HotStart ready mix (Kapa Biosystems, Wilmington, MA). For post library PCR, we used AMPure XP beads at a 1× ratio and optimized pH and salt concentration to maintain size distribution of the library.

To improve visualization, size selection was performed after post library PCR on a 2% agarose gel. The region covering 125–1000 bp was excised and gel residue was removed using the QIAquick Gel Extraction Kit (Qiagen). After final quality control on a BioAnalyser 2100 using a DNA 1000 chip (Agilent, Santa Clara, CA), the 4C libraries were used for flowcell cluster formation on a cluster station and 36 bp single end sequencing was performed on a Genome Analyzer IIx.

Sequence data were scanned for the respective primer/restriction site pairs associated with the region of interest. Any sequences not matching perfectly these criteria were discarded. The remaining sequences were trimmed to remove the primer and then mapped to the *Saccharomyces cerevisiae* genome using PatMaN (*Prüfer et al., 2008*). This was performed in two rounds. Firstly, aligning with no mismatches, removing the matching sequences from the input file and then aligning again with 1 mismatch. The two sets of PatMaN results were then merged for each genomic location and then counts were produced for each position.

## Microscopy

Images of live budding yeast cells containing two differentially fluorescently marked loci at 144 kb distance from each other on the long arm of chromosome 5 (*Rohner et al., 2008*) were acquired using a DeltaVision Olympus IX70 inverted microscope with a 100× (NA = 1.40) PlanApo objective. 100 pairs of two independent biological repeats, each, were scored. Distances between the brightest centers of the two fluorescent foci were measured in 3 dimensional reconstructions of the cells using Softworx (DeltaVision, GE Healthcare, Pittsburgh, PA).

## Theory and functions of the Type I and II model

For each simulation, a virtual chromosome of ~300 kb is constructed. Along this length, 2000 nucleosomes are represented as spheres of 10 nm diameter, with condensin binding sites placed at ~10 kb intervals. The distance between the centers of each consecutive nucleosome bead is set to 15 nm, with the interconnection between them modeled as a spring.

The conformation of a self-avoiding chromosome structure used in each Brownian dynamics simulation is obtained in two steps. Firstly, a 3D Hilbert curve filling the space of a 150 nm length cube is generated. Secondly, using Brownian dynamics this compacted chain is allowed to expand to a cylinder of diameter 200 nm and of unlimited length. After this, spatial constraints are removed to obtain a chromatin chain topology of a relaxed interphase state. Brownian dynamics simulation is run until a steady state of the chromosome compaction ratio is achieved (as shown in *Figure 2C* where the compaction of chromosome stabilized from ~5 min onwards).

The Type I and II models are conceptually similar to a dynamic loop model (*Zhang and Heermann, 2011*), in a way that genomically distant sections of the chromatin fibre can cross-link for a fixed amount of time, as facilitated by condensin molecules, when they come into physical proximity of each other. An important feature of our Type I and II model is the probabilistic nature of the cross-linking mechanism, which allows the organization of the chromatin fibre to be dynamic, rather than being a fixed structure. Another important feature of the Type I and II models is the application of physical forces at the nucleosome level. The trajectory of the fiber backbone is regulated by forces, as described below, based on first principle physics. The balance of all forces is calibrated based on experimental measurements of nucleosome movements and the angles between neighboring nucleosome linkers. Without arbitrary parameters to pre-define the conformation of the chromosome chain (i.e., no conformation parameters), the models allow a direct examination of how chromosome conformations emerge from different modes of condensin interactions (i.e., stochastic pairwise interactions and higher order condensing assemblies for the Type I and II models, respectively).

### Forces employed

In both the Type I and II models, the movement of a specific bead $i$ during simulation is regulated by a summation of forces, $\overrightarrow{F}_i^{all}$:

$$\overrightarrow{F}_i^{all} = \overrightarrow{F}_i^{entropic} + \overrightarrow{F}_i^{tension} + \overrightarrow{F}_i^{repulsion} + \overrightarrow{F}_i^{attraction} + \overrightarrow{F}\left(d_{i,j}, p\right)_i^{condensin}, \tag{1}$$

where $\overrightarrow{F}_i^{entropic}$ is an entropic force for diffusion-based movements of nucleosome beads; $\overrightarrow{F}_i^{tension}$ is a tension force originated from the nucleotide linkers joining two neighboring beads; $\overrightarrow{F}_i^{repulsion}$ is a repulsion force to avoid clashes between bead $i$ and other beads within its vicinity; $\overrightarrow{F}_i^{attraction}$ is a weak force applied between beads $i$ and $i \pm 2$ to regulate the distribution of the α angle (the angle between the DNAs emerging from the surface of nucleosomes); $\overrightarrow{F}(d_{i,j}, p)_i^{condensin}$ is a tension force originating from the bonds connecting two condensin binding sites $i$, $j$. It is applied when bead $i$ represents a condensin binding site and when beads $i$, $j$ are within 40 nm of each other ($d_{i,j} \leq 40$ nm). The $p$ parameter represents the dissociation probability of condensin molecules for regulating the maintenance of $\overrightarrow{F}(d_{i,j}, p)_i^{condensin}$. Algorithmically, the probability $p$ is implemented through a random number generated at each time step for each $\overrightarrow{F}(d_{i,j}, p)_i^{condensin}$: if a random number for a $\overrightarrow{F}(d_{i,j}, p)_i^{condensin}$ between a pair of interacting condensin sites is less than a certain threshold $p$, then $\overrightarrow{F}(d_{i,j}, p)_i^{condensin}$ becomes zero.

The only difference between the Type I and Type II model is in $\overrightarrow{F}(d_{i,j}, p)_i^{condensin}$: In the Type I scenario, the force is applied between condensin site $i$ and *one* of its spatially nearby condensin sites, whereas in the Type II model up to *two* of the nearby condensin sites are regulated by the force. More details of the $\overrightarrow{F}(d_{i,j}, p)_i^{condensin}$ are described in its section below.

## $\overrightarrow{F}_i^{entropic}$: entropic force

An entropic force $\overrightarrow{F}_i^{entropic}$ is applied to each bead, at each simulation step, in the following form:

$$\overrightarrow{F}_i^{entropic} = c_1 \times \overrightarrow{u}, \tag{2}$$

where $c_1$ is a constant determining the magnitude of the force and $\overrightarrow{u}$ is a vector specifying its direction. The value of $c_1$ allows the bead to have an average movement based on the friction coefficient of solvated nucleosomes (*Robert, 1995*), on a scale consistent with experimental observations. Nucleosome bead movement is principally regulated by the entropic force and the spring constant of the nucleosome linker (see '$\overrightarrow{F}_i^{tension}$: tension force', below). This parameter pair was chosen such that the nucleosome displacement distribution over short (30 ms) time intervals was compatible with that observed in mammalian interphase cells (*Hihara et al., 2012*), to our knowledge only currently available experimental measurement of this distribution. In our model, using the parameter set given below, the displacement distribution is of comparable shape to the experimental observation, with a mean and standard deviation of 30.9 $\pm$ 13.4 nm/30 ms (Type I model) or 30.8 $\pm$ 13.3 nm/30 ms (Type II model), compared to a mean of 51.2 nm/30 ms in the experiment (*Figure 1—figure supplement 1*). Energy from numerous chromosomal processes is expended in the human interphase nucleus, which could cause a somewhat greater mobility of nucleosomes compared to our model that is restricted to Brownian motion. Key aspects of our model were robust to the exact values of this parameter pair (see 'Model robustness', below).

## $\overrightarrow{F}_i^{tension}$: tension force

Here, we approximate a nucleotide linker, joining two neighboring beads, as an elastic spring following Hooke's law:

$$\overrightarrow{F}_i^{tension} = K_s \times (d_{i,i+1} - c_2) \times \hat{u}_{i,i+1} + K_s \times (d_{i,i-1} - c_2) \times \hat{u}_{i,i-1}, \tag{3}$$

where $K_s$ is the spring constant of the nucleotide linker, $d_{i,j \pm 1}$ is the distance between the center of two neighboring beads $i$ and $j$; $c_2$ is a constant describing the natural (non-stretched or non-compressed) length of the nucleotide linker; $\hat{u}_{i,i+1}$, $\hat{u}_{i,i-1}$ are unit vectors determining the direction of the force (from bead $i$ to its neighboring bead $i + 1$ and $i - 1$, respectively).

## $\overrightarrow{F}_i^{repulsion}$: repulsion force

To avoid overlaps between beads, a distance-dependent repulsive force between two beads that are within 15 nm of each other is applied:

$$\begin{cases} \overrightarrow{F}_i^{repulsion} = c_3 \times \hat{u}_{i,j}, \quad d_{i,j} < 10 \text{ nm} \\ \overrightarrow{F}_i^{repulsion} = \dfrac{c_3 \times \hat{u}'_{i,j}}{d_{i,j}^{12}}, \quad 10 \leq d_{i,j} \leq 15 \text{ nm} \end{cases}, \tag{4}$$

where $c_3$ is the magnitude of repulsion force. The direction of the force is given by $\hat{u}_{i,j}$ and $\hat{u}'_{i,j}$, dimensionless and dimensionful unit vectors from bead $i$ to $j$, respectively; $d_{i,j}$ is the distance between two nearby beads $i$ and $j$. The overall shape of the function resembles the shape of soft-core Van der Waals forces (with the scaling parameter $\alpha$ closer to 1 than 0), as typically applied in molecular dynamics simulation force fields (**Ponder and Case, 2003**).

### $\overrightarrow{F}_i^{attraction}$: attraction force

A weak attractive force is applied between beads $i$ and $i \pm 2$:

$$\overrightarrow{F}_i^{attraction} = c_4 \times \hat{u}_{i,i+2} + c_4 \times \hat{u}_{i,i-2}, \tag{5}$$

where $c_4$ is the magnitude of the force, proportional to the distance between beads $i$, $i + 2$ and between beads $i$, $i - 2$, respectively. $\hat{u}_{i,i+2}$ and $\hat{u}_{i,i-2}$ are dimensionless unit vectors determining the direction of the force (from bead $i$ to $i + 2$ and from bead $i$ to $i - 2$, respectively). The force fine-tunes the distribution of the $\alpha$ angle between three consecutive beads and does not serve as a driving factor for any pre-defined local structures: the $\alpha$ angle distribution for our simulated chromosomes has a bell shape curve, which peaks at around 70° (benchmarked with experimental measurements of the $\alpha$ angle of 75°, based on the nucleosome crystal structure [**Luger et al., 1997**]) and tails off at around 20° and 180°, consistent with the relative flexibility that is thought to characterize the $\alpha$ angle (**Figure 1—figure supplement 2**) (**Bednar et al., 1998**; **Engelhardt, 2007**). This means that the model does not exclude the possibility of forming small fragments of locally compact structures that resemble the 20–30 nm fibers widely speculated in other models (the $\alpha$ angle of a 30 nm fiber is centered at 36° or 108°). Although the majority of the simulated chromosomes have a broad range of local conformations as can be seen in the supplementary videos.

### $\overrightarrow{F}(d_{i,j}, p)_i^{condensin}$: condensin tension force

This force maintains the bond between two condensin binding sites that are within 40 nm of each other. Here, two different forms of $\overrightarrow{F}(d_{i,j}, p)_i^{condensin}$ are applied in the Type I (**Equation 6.1, 6.3, 6.4**) and Type II (**Equation 6.2, 6.3, 6.4**) models. Both model the bond between condensin binding sites as an elastic spring following Hooke's law:

$$\overrightarrow{F}(d_{i,j}, p)_i^{condensin} = K_{condensin} \times (d_{i,j} - c_5) \times \hat{u}_{i,j}, \tag{6.1}$$

$$\overrightarrow{F}(d_{i,j}, p)_i^{condensin} = K_{condensin} \times (d_{i,j} - c_5) \times \hat{u}_{i,j} + K_{condensin} \times (d_{i,k} - c_5) \times \hat{u}_{i,k}, \tag{6.2}$$

$$\overrightarrow{F}(d_{i,j}, p)_i^{condensin} = 0, \text{ when } d_{i,j} > 40 \text{ nm}, \tag{6.3}$$

$$\overrightarrow{F}(d_{i,j}, p)_i^{condensin} = 0, \text{ when } p \text{ is not satisfied}, \tag{6.4}$$

where in both Type I and II models $K_{condensin}$ is the spring constant of the bond between two interacting condensin binding sites; $d_{i,j}$ is the distance between the center of two condensin sites $i$ and $j$; $p$ is the dissociation probability of condensin molecules; $c_5$ is the average distance maintained between two nearby condensin sites; $\hat{u}_{i,j}$ is an unit vector determining the direction of the force (from condensin site $i$ to condensin site $j$). In case of the Type II model, $d_{i,k}$ is the distance between the center of condensin site $i$ and an additional site $k$; $\hat{u}_{i,k}$ is a unit vector determining the direction of this additional force from condensin site $i$ to condensin site $k$. Here we define $c_5$ as 30 nm, shorter than the distance threshold used for determining a bond maintained between two nearby condensin sites (40 nm). This allows two nearby condensin sites to re-form the bond if they are still within a 40 nm distance after the initial interaction bond between them breaks (monitored during a window of three simulation time steps).

In the Type I model, when a condensin binding site has more than one nearby condensin sites within the 40 nm range, only one interaction with one of the nearby sites is allowed based on a random choice. In the Type II model, when a condensin binding site has more than two nearby condensin sites within the 40 nm range, only two interactions can be formed based on random choices.

## The mid-point scheme employed for calculating the movement of beads

At each simulation step, the movement of each bead is calculated through the following mid-point numerical scheme:

$$(x_i)_{\frac{1}{2}} = (x_i)_0 + \frac{\delta t}{2} \times \left( \left( \left(\overrightarrow{F}_i^{entropic}\right) + \left(\overrightarrow{F}_i^{tension}\right) + \left(\overrightarrow{F}_i^{repulsion}\right) + \left(\overrightarrow{F}_i^{attraction}\right) + \left(\left(\overrightarrow{F}(d_{i,j},p)_i^{condensin}\right)\right) \right)_0 \right),$$

$$(x_i)_1 = (x_i)_0 + \delta t \times \left( \left( \left(\overrightarrow{F}_i^{entropic}\right)_0 + \left(\overrightarrow{F}_i^{tension}\right)_{\frac{1}{2}} + \left(\overrightarrow{F}_i^{repulsion}\right)_{\frac{1}{2}} + \left(\overrightarrow{F}_i^{attraction}\right)_{\frac{1}{2}} + \left(\left(\overrightarrow{F}(d_{i,j},p)_i^{condensin}\right)\right)_{\frac{1}{2}} \right) \right),$$

$$(7)$$

where $(x_i)_{\frac{1}{2}}$ and $(x_i)_1$ are the positions of a bead at the mid and end-point of each simulation time step, respectively; $\delta t$ is the *in silico* simulation time step, $(\overrightarrow{F}_i)_0$ and $(\overrightarrow{F}_i)_{\frac{1}{2}}$ are the forces attributed to each bead at simulation time steps $\delta t$ and $\delta t/2$, respectively.

## List of parameters regulating the bead movement

| Parameters | Values ('*': non-defined values) | Dimension ('-': dimensionless) | Host function |
|---|---|---|---|
| $c_1$ | 24.5 | pN | $\overrightarrow{F}_i^{entropic}$ |
| $\overrightarrow{u}$ | $1 \leq |\overrightarrow{u}| \leq -1$ | - | $\overrightarrow{F}_i^{entropic}$ |
| $K_s$ | 50 | pN nm$^{-1}$ | $\overrightarrow{F}_i^{tension}$ |
| $d_{i,i\pm1}$ | * | nm | $\overrightarrow{F}_i^{tension}$ |
| $C_2$ | 15 | nm | $\overrightarrow{F}_i^{tension}$ |
| $\hat{u}_{i,i\pm1}$ | * | - | $\overrightarrow{F}_i^{tension}$ |
| $C_3$ | 10 | pN | $\overrightarrow{F}_i^{repulsion}$ |
| $\hat{u}_{i,j}$ | * | - | $\overrightarrow{F}_i^{repulsion}$ |
| $\hat{u}'_{i,j}$ | * | nm | $\overrightarrow{F}_i^{repulsion}$ |
| $d_{i,j}$ | * | nm | $\overrightarrow{F}_i^{repulsion}$ |
| $c_4$ | * | pN | $\overrightarrow{F}_i^{attraction}$ |
| $\hat{u}_{i,i\pm2}$ | * | - | $\overrightarrow{F}_i^{attraction}$ |
| $K_{condensin}$ | 50 | pN nm$^{-1}$ | $\overrightarrow{F}(d_{i,j},p)_i^{condensin}$ |
| $d_{i,j}$ | * | nm | $\overrightarrow{F}(d_{i,j},p)_i^{condensin}$ |
| $P$ | G1 phase: $10^{-3}$ M phase: $10^{-4}$ | - | $\overrightarrow{F}(d_{i,j},p)_i^{condensin}$ |
| $C_5$ | 30 | nm | $\overrightarrow{F}(d_{i,j},p)_i^{condensin}$ |
| $\hat{u}_{i,j}$ | * | - | $\overrightarrow{F}(d_{i,j},p)_i^{condensin}$ |

## Model robustness

To estimate the robustness of the difference between the Type I and Type II models, we have sampled the spring constant $K_s$ and entropic force $\overrightarrow{F}_i^{entropic}$ values, the most important parameters that regulate the movement of the nucleosome beads. We searched for combinations of $K_s$ and $\overrightarrow{F}_i^{entropic}$ that allow nucleosome beads to exhibit movements at rates compatible with those measured by *Hihara et al. (2012)*. We plotted the relationship between $K_s$ and $\overrightarrow{F}_i^{entropic}$ values, and selected a point ($K_s = 10$ pN/nm and $\overrightarrow{F}_i^{entropic} = 11$ pN) that generated a low average tension between consecutive nucleosomes (2.2 pN) to compare with our original model based on the above tabulated parameters

($K_s$ = 50 pN/nm and $\overrightarrow{F}_i^{\,entropic}$ = 24.5 pN) that result in an average inter-nucleosome tension of 4.7 pN. Simulations with these new parameter values using the Type I model resulted in similar chromosome dimensions and native-like condensation, while the Type II model again showed a tendency to overly compact chromosomes. The fraction of long-range interactions beyond 100 kB was somewhat lower but compatible with the experimental data in the Type I model. They markedly exceeded the observed range in the Type II model. We conclude that the main features and differences between the Type I and Type II models are insensitive to variations in the $K_s$ and $\overrightarrow{F}_i^{\,entropic}$ values. We note that the inter-nucleosome tension with either parameter pair is insufficient to disrupt the histone/DNA interaction, though forces in the low pN range may in some cases, depending on the DNA sequence, lead to elastic DNA breathing on the nucleosome surface (*Ngo et al., 2015*).

## Mapping the in silico time to in vivo time

Here, we define the *in silico* time step $\delta t$ as follows, taking into account the diffusion of solvated nucleosomes (*Grassia and Hinch, 1996*):

$$\delta t = \frac{\hat{t} \times D}{\lambda^2}, \tag{8}$$

where $\hat{t}$ is in vivo time, $\lambda$ is the natural length of a nucleotide link joining two neighbouring beads (set as 15 nm in our system); $D$ is the diffusion coefficient of a nucleosome, which related to the friction coefficient, $f$ as:

$$D = \frac{k_B \times T}{f}, \tag{9}$$

where $k_B$ is the Boltzmann constant, $T$ is the temperature.

The value of $f$ for solvated nucleosomes is approximately $3 \times 10^{-7}$ g s$^{-1}$ (*Robert, 1995*). Hence the in vivo time corresponding to each *in silico* step at a temperature of 25°C is approximately:

$$\hat{t} = \frac{\delta t \times \lambda^2}{\left(\frac{k_B \times T}{f}\right)} = \frac{1 \times 15^2 \text{nm}^2}{1.33 \times 10^7 \text{s g}^{-1} \text{ pN nm}} = 1.69 \times 10^{-5} \text{s}. \tag{10}$$

Our simulations take approximately 25 million simulation steps to reach a new compaction steady state after a change in dissociation probability of condensin interactions, which corresponds to approximately 7 min in vivo. This is compatible with requirements for the in vivo chromosome condensation in budding yeast.

## K and persistence length determination of the simulated chromatin chain

We calculated the Pearson's definition of K, defined as the fourth central moment divided by the square of the variance, using the following function:

$$K = \frac{\mu_4}{\sigma^4}, \tag{11}$$

where $\mu_4$ is the fourth moment about the mean and $\sigma$ is the standard deviation.

We applied an orientational correlation function, $K_{or}$, to estimate the persistence length, $L_p$, at different genomic distances (*Strobl, 1997*). $K_{or}$ gauges the rigidity of a polymer by the orientational correlation of pairs of locations. Let u(L) be a tangent unit vector describing the direction of the chain at contour length L. The correlation between two points separated by contour length L′ is:

$$u(L) \cdot u(L + L'). \tag{12}$$

The $K_{or}$ value is the average of correlations, considering all pairs of tangent vectors separated by contour distance L′, extracted from an ensemble of equilibrated structures:

$$K_{or} = \langle u(L) \cdot u(L + L') \rangle. \tag{13}$$

Eventually the $L_p$ value is estimated as the integral width of the $K_{or}$ function:

$$L_P = \int_0^\infty K_{or}(L')d(L'). \tag{14}$$

We note that this persistence length determination applies to a loop-based chromatin model (*Zhang and Heermann, 2011*). Loops constrain free movement of the chromatin fibre and, unlike in a worm-like model, $L_p$ increases with greater genomic distances.

## Approximation of the budding yeast genome volume based on the simulation results

We estimated the volume of the 300 kb chromosome chain in our simulations through gridded spaces. Firstly, we defined a space big enough to enclose the chromosome then we partitioned the space into joined 3D grids. We counted the number of grid cells occupied by at least one nucleosome. The volume of the chromosome was the sum of the volumes of the occupied grid cells. The accuracy of this approach depends on the grid size used to partition the space. Small grids underestimate the chromosome volume by omitting internal spaces that are surrounded by the chromosome and that are hence not reachable by other chromosomes. Large grids tend to overestimate the volume by including an excess of external surrounding space that could be available to neighboring chromosomes. Here, we use a grid of 50 nm cubes, corresponding to the observed persistence length at small distances in our simulated structures. As the chromosome volume is determined by the curvature of the chromatin chain, closely related to its persistence length, this is aimed to reduce the effect of both over- or underestimating chromosome volume.

Using this approach, the estimated volumes of Type I and Type II chromosomes in interphase were $5.0 \pm 0.17 \times 10^{-2}$ μm$^3$ and $4.8 \pm 0.3 \times 10^{-2}$ μm$^3$, respectively, based on the 60 sampled structures. By scaling both numbers by a factor of 40 we project the volume of our 300 kb chromosome to the volume of the 12 Mb budding yeast genome. This yields approximately $2 \pm 0.07$ μm$^3$ for chromosomes that behave according to the Type I model. This accounts for half the volume of the yeast nucleus, assuming a diameter of 2 μm and corresponding volume of approximately 4 μm$^3$ (*Gasser, 2002*). This suggests that the budding yeast nucleus comfortably accommodates the yeast genome based on the size of our simulated chromosome structures.

## Acknowledgements

We thank Ozge Kurgcuoglu and Alexander Tournier for their contributions at the outset of the study, Susan Gasser for reagents and for generously sharing primary data, Kazuhiro Maeshima, John Marko, Adam Thorn and members of our laboratories for discussions and critical reading of the manuscript.

## Additional information

### Funding

| Funder | Grant reference | Author |
|--------|-----------------|--------|
| Cancer Research UK | Postdoctoral Fellowship | Tammy MK Cheng |
| European Research Council | Advanced Grant | Tammy MK Cheng, Frank Uhlmann |
| EMBO | Long Term Fellowship | Sebastian Heeger |
| Academia Sinica | | Jon Wright, Carmay Lim |
| Ministry of Science and Technology, Taiwan (MOST) | | Carmay Lim |

The funders had no role in study design, data collection and interpretation, or the decision to submit the work for publication.

## Author contributions

TMKC, Conception and design, Acquisition of data, Analysis and interpretation of data, Drafting or revising the article; SH, Acquisition of data, Analysis and interpretation of data; RAGC, JW, Performed coding and programming; NM, Conducted high throughput sequencing; AS, Bioinformatics analysis; CL, Made infrastructure available; PAB, FU, Conception and design, Drafting or revising the article

# Additional files

## Major dataset

The following dataset was generated:

| Author(s) | Year | Dataset title | Dataset ID and/or URL | Database, license, and accessibility information |
|---|---|---|---|---|
| Cheng TMK, Heeger S, Chaleil RAG, Matthews N, Stewart A, Wright J, Lim C, Bates PA, Uhlmann F | 2015 | Data from: A simple biophysical model emulates budding yeast chromosome condensation | 10.5061/dryad.78622 | Available at Dryad Digital Repository under a CC0 Public Domain Dedication. |

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
