## [Decision Letter]

Thank you for sending your work entitled “A simple biophysical model emulates budding yeast chromosome condensation” for consideration at *eLife*. Your article has been favorably evaluated by Aviv Regev (Senior editor) and two reviewers, one of whom is a member of our Board of Reviewing Editors.

The Reviewing editor and the other reviewer discussed their comments before we reached this decision, and the Reviewing editor has assembled the following comments to help you prepare a revised submission.

This manuscript takes a computational approach to study the mechanisms responsible for chromosome condensation in budding yeast. Using the simplifying assumption of a bead spring model for nucleosomes and various ways to represent condensin (bridging two sites or more sites, Type I and Type II) they go a remarkable long way toward capturing the major experimental measurement in the literature as well as their new data. Known sites of condensin binding are assumed to coalesce into one complex temporarily if they come within 40 nm of each other, the capture radius consistent with the physical dimension of condensin. The lifetime of the complex is used as a parameter that can be used to tune the dynamic nature of such interaction and is the sole parameter varied between interphase and mitotic chromatin. In Type II model, more than two condensin binding sites are allowed to coalesce. Remarkably, simulations can explain the main features of the experimental data with Type I model performing significantly better. What's attractive about this work is that a simple physical model can get us quite far in explaining the chromosome behavior at a quantitative level, and that the model is general enough that straightforward extensions to include other effects may be possible in the future. In general this is a nice manuscript and clearly explained.

Main points:

1) It is unclear how some of the parameters were set. Presumably, these parameters are determined to satisfy the constraints provided by modern biophysical measurements referenced but currently, their description is too cursory to help the readers understand why these numbers were chosen. For example, entropic force are set at 17 pN, presumably to give a kick to a nucleosome at each time point, but it is not clearly stated why previous observations referred to demand this particular value. The same for the spring constant for internucleosomal tension, 50 pN/nm. For a typical displacement during the simulation, what is the resulting internucleosomal tension? Please note that if the forces go beyond a few pN, the nucleosomal DNA will unravel to a large degree, which may not be physiological, and may also affect the simulation itself to such a degree that required modifications to the model. If there is not actual basis for the number, does it mean that the authors have tried a range of forces and spring constants and only this set of values is able to fit the data? What if the Type II model works better if a different set is used for this parameter?

There are two sentences hinting at how the parameters were chosen.

“The balance of all forces are calibrated based on experimental measurements of nucleosome movements and the angles between neighboring nucleosome linkers.”

“Parameters with defined values are benchmarked based on the observed α angles and local movements of nucleosomes, as described above.”

But there isn't even any reference given what these experimental measurements of nucleosome movements are and the alpha angles. At the minimum, they need to show how the chosen set of parameters are compared to the experimental measurements of nucleosome movements and the angles and show that these choices are unique. If there are other sets of parameters that are similarly good, they need to show that the main conclusions of the paper still hold with those alternative sets of parameters.

2) They find the measured distance using TetO and LacO markers (670 nm) to match the starting configuration of the model. The statement that the correspondence implies the configurations are similar is over-interpreted. These data are correlative, not proof of a common structural basis. In addition, they measured only one set of loci, which limits the statistical power of the statement quoted below:

“This striking correspondence implies that interphase chromatin in vivo adopts a configuration of similar dimensions to an unconstrained nucleosome fiber.”

3) The authors state that persistence length increase with greater genomic distance, as expected from loop models. As the authors know, Lp is defined as <cosΘ(s)> = e(-s/l), where Θ(s) is the angle between two ends on a chain separated by contour length s. My understanding was that the persistence length was dictated by the physical properties of the chain, independent of where you are on the chain. Again, there is no question that the chain explores more space in the middle (Increased Rc, see Verdaasdonk, ibid.) and less at the ends, but this is not due to change in Lp, rather at least for a chromosome is a result from tethering. The authors need to justify their statement that Lp increases with increased genomic distance.

4) The conclusion that condensin is responsible for individualization is unfortunately rather trivial. It is based on the fact that intramolecular interactions will always dominate intermolecular interactions. This falls out of the polymer books referred to above. Thus it is the case that condensin will further the tendency for chromosome individualization, but it is not causative.

5) One suggestion that is hinted at in the Introduction but is not well emphasized is the proposal that no scaffolding structures are required to account for the experimental data. In this sense, [35] was referred to as being supportive of the loops with scaffolding model, but in fact [35] was agnostic about the presence or absence of scaffolding (see Figure 3 of [35]). Would it be possible to suggest that the present work is able to differentiate between models that [35] could not? On a similar note, 2014 Cell paper by the Heard lab should be referenced as it examined the chromosomal conformations using a physics-based polymer model.

6) Is the inter-chromatin marker distance determined via imaging two colors a distance projected to the plane of imaging or a 3D distance? The method indicates that it is the 3D distance. If so, it should be mentioned explicitly in the main text. Also, determining distance based on two color imaging is not trivial due to chromatic aberration and the authors need to elaborate how much confidence they have on the distribution of distances. This is critical because the deviation of the distribution from gaussian was used to deduce Kurtosis and to provide additional support for their model.

In summary, this is an interesting model that emphasizes how few parameters are needed to recapitulate the course-grain behavior of eukaryotic chromosomes. There are several places that the authors need to clarify and expand to make the paper more accessible to beginners in polymer physics. The main concern is about the choice of parameters and there isn't information given how the parameters were optimized to fit what types of experimental constraints, and how unique the parameters are. What if another set of parameters that can fit the constraints just as well show that the main conclusions do not hold anymore?

[Editors' note: further revisions were requested prior to acceptance, as described below.]

Thank you for sending your work entitled “A simple biophysical model emulates budding yeast chromosome condensation” for consideration at *eLife*. Your revision has been evaluated by a member of our Board of Reviewing Editors. Although the revision has improved the manuscript significantly, it needs additional revision to address one major point and one minor point described below.

Major point: The authors still do not show how their parameters are benchmarked.

“Parameters with defined values are benchmarked based on experimentally observed α angles (Luger, 1997; Bednar, 1998; Engelhardt, 2007) and local movements of nucleosomes (Hihara, 2012), as described above.”

At the minimum, they need to show the experimental observed alpha angle distribution, i.e. probability distribution vs alpha and show that their simulated alpha angle distribution matches the experimental distribution. In addition, they need to show, preferentially in figures, how the simulated local movements of nucleosomes match the experimental movements.

Minor point: They argue in the rebuttal that 2-5 pN of force should not unravel DNA by referencing a 2001 paper. However, more recent papers in PNAS by Michelle Wang's group and Carlos Bustamante's group showed that DNA indeed unravels at such forces.

---

## [Author Response]

1) It is unclear how some of the parameters were set. Presumably, these parameters are determined to satisfy the constraints provided by modern biophysical measurements referenced but currently, their description is too cursory to help the readers understand why these numbers were chosen. For example, entropic force are set at 17 pN, presumably to give a kick to a nucleosome at each time point, but it is not clearly stated why previous observations referred to demand this particular value. The same for the spring constant for internucleosomal tension, 50 pN/nm. For a typical displacement during the simulation, what is the resulting internucleosomal tension? Please note that if the forces go beyond a few pN, the nucleosomal DNA will unravel to a large degree, which may not be physiological, and may also affect the simulation itself to such a degree that required modifications to the model. If there is not actual basis for the number, does it mean that the authors have tried a range of forces and spring constants and only this set of values is able to fit the data? What if the Type II model works better if a different set is used for this parameter?

*There are two sentences hinting at how the parameters were chosen*:

*“The balance of all forces are calibrated based on experimental measurements of nucleosome movements and the angles between neighboring nucleosome linkers*.*”*

*“Parameters with defined values are benchmarked based on the observed α angles and local movements of nucleosomes, as described above*.*”*

*But there isn't even any reference given what these experimental measurements of nucleosome movements are and the alpha angles. At the minimum, they need to show how the chosen set of parameters are compared to the experimental measurements of nucleosome movements and the angles and show that these choices are unique. If there are other sets of parameters that are similarly good, they need to show that the main conclusions of the paper still hold with those alternative sets of parameters*.

The reviewers raise an important point regarding parameter choice and robustness of our model, especially regarding the distinction between the Type I and Type II models. We have addressed this concern in two ways. Firstly, we refer to the experimental studies more prominently, that we used to benchmark the key parameters referred to by the reviewers (Luger, 1997; Bednar, 1998; Engelhardt, 2007 for the alpha angle and Robert, 1995 and Hihara, 2012 for the entropic force; please see Materials and methods, subsection headed “Theory and functions of the Type I and II model”).

More importantly, we have performed a new series of simulations in which we have explored the parameter space for the two most important parameters that regulate the movement of nucleosome beads, the spring constant K_s_ and the entropic force. The results of these analyses are summarized in a new paragraph “Model robustness”.

In brief, we repeated our simulations with a parameter pair as distant from our original parameter pair as possible, while retaining nucleosome movement characteristics compatible with the constraints from the experimental observations by Hihara et al. 2012. This led us to a soft spring constant and lower entropic force (K_s_ = 10 pN/nm and F^entropic^ = 11 pN), compared to (K_s_ = 50 pN/nm and F^entropic^ = 24.5 pN) in the original model. This results in a weakening of the tension between consecutive nucleosomes to ∼2.2 pN (compared to ∼4.7pN using the original parameters). Of note, tension in either range should not lead to substantial DNA unwrapping from the nucleosomes which occurs when tension approaches 20 pN (Bennik et al. 2001). The analysis of the resulting simulations showed that the main features and differences between the Type I and Type II are similar to what is observed with the original parameters and therefore that the model is robust.

2) They find the measured distance using TetO and LacO markers (670 nm) to match the starting configuration of the model. The statement that the correspondence implies the configurations are similar is over-interpreted. These data are correlative, not proof of a common structural basis. In addition, they measured only one set of loci, which limits the statistical power of the statement quoted below:

*“This striking correspondence implies that interphase chromatin in vivo adopts a configuration of similar dimensions to an unconstrained nucleosome fiber*.*”*

We agree with the reviewers that the results shown in Figure 2 by themselves do not allow a conclusion about the similarity of chromatin configurations between observation and model. We did not intend to make such an assertion; rather we concluded that the dimensions of the respective configurations are comparable. To preempt any possible misunderstanding, we have rephrased this sentence to read:

“This striking correspondence suggests that interphase chromatin in vivo adopts a configuration of similar dimensions as compared to the dimensions of an unconstrained nucleosome fiber.”

*3) The authors state that persistence length increase with greater genomic distance, as expected from loop models*. *As the authors know, Lp is defined as <cosΘ(s)> = e(-s/l), where Θ(s) is the angle between two ends on a chain separated by contour length s. My understanding was that the persistence length was dictated by the physical properties of the chain, independent of where you are on the chain. Again, there is no question that the chain explores more space in the middle (Increased Rc, see Verdaasdonk, ibid.) and less at the ends, but this is not due to change in Lp, rather at least for a chromosome is a result from tethering. The authors need to justify their statement that Lp increases with increased genomic distance.*

The persistence length definition referred to by the reviewers (K_or_ defined as <cos(Θ)>·= exp(-s/L_p_)) can be derived from the generic description, given by our equation (Naumova, 2013), for the special case of a worm-like chromatin model. However, the use of this equation does not yield a good fit to our data. The reason is that in a loop-based model, unlike in a worm-like model, the movement of the chromatin fibre is constrained by the loops, such that Lp increases with greater genomic distances. This becomes apparent when analyzing our chromosome structures using equation (Naumove, 2013), but also when comparing persistence length measurements in budding yeast that were performed at different length scales (e.g. compare [11] and [25]_). This is clarified in the Results section (subsection “Simulated and experimental polymer characteristics”), as well as in an explanatory note to the Methods section (“Kurtosis and persistence length determination of the simulated chromatin chain”). We also include an additional reference that has used equation (Naumova, 2013) to determine the persistence length of a loop-based chromatin polymer chain (Zhang et al. 2011).

4) The conclusion that condensin is responsible for individualization is unfortunately rather trivial. It is based on the fact that intramolecular interactions will always dominate intermolecular interactions. This falls out of the polymer books referred to above. Thus it is the case that condensin will further the tendency for chromosome individualization, but it is not causative.

The reviewers are right that chain individualization is intrinsic to polymer behavior. However, our analysis goes further than merely demonstrating that condensin furthers this intrinsic tendency. We compare model behavior using two modes of condensin action, Type I or Type II. This leads to the realization that only Type I condensin action efficiently promotes individualization. Type II action often results in permanent entanglements. We would like to argue that the computational analysis of this behavior provides valuable insight and constrains the way in which condensin is likely to act.

5) One suggestion that is hinted at in the Introduction but is not well emphasized is the proposal that no scaffolding structures are required to account for the experimental data. In this sense, [35] was referred to as being supportive of the loops with scaffolding model, but in fact [35] was agnostic about the presence or absence of scaffolding (see Figure 3 of [35]). Would it be possible to suggest that the present work is able to differentiate between models that [35] could not? On a similar note, 2014 Cell paper by the Heard lab should be referenced as it examined the chromosomal conformations using a physics-based polymer model.

We thank the reviewers for suggesting that we give more attention to the question of a chromosome scaffold. [35] is an overview article that indeed remains agnostic as to the presence or absence of a chromosome scaffold. The reference discusses hierarchical folding, which often has been proposed as part of chromosome scaffold models. Figure 6 in this reference shows axial condensin staining, which has often been used as an argument to support the idea of a scaffold. In this sense the reviewers are right that our model allows us to differentiate between models that [35] could not. We have therefore rephrased a sentence in our Discussion: “These results suggest that interphase chromatin exists in a relatively relaxed state inside a budding yeast nucleus and that there may be no need to invoke major forces, scaffolds or other organizing principles to explain its packing”.

Furthermore, we fully agree that the 2014 Cell paper from the Heard lab is relevant for our discussion, as it uses a physics-based polymer model, albeit at low resolution, to explore topologically associating domains on a mammalian X-chromosome. We now include this reference in our Discussion: “Other models were aimed at describing whole genome organization in the yeast nucleus, or mammalian chromosome behavior at lower resolution (references including [19]).”

*6) Is the inter-chromatin marker distance determined via imaging two colors a distance projected to the plane of imaging or a 3D distance? The method indicates that it is the 3D distance. If so, it should be mentioned explicitly in the main text. Also, determining distance based on two color imaging is not trivial due to chromatic aberration and the authors need to elaborate how much confidence they have on the distribution of distances. This is critical because the deviation of the distribution from gaussian was used to deduce Kurtosis and to provide additional support for their model*.

We clarify in the main text that we have indeed measured in 3 dimensions the distance between the chromatin markers (please see the sentence “this allowed us to measure their in vivo 3D distance…”).

In addition, the reviewers’ point of caution about two color distance measurements due to chromatic aberrations is well taken. We have ensured that distance measurements were only made close to the center of the lightpath, where chromatic aberration is minimal. With this in mind, we do believe that the two color observations give us more accurate distance measurements, especially at the important shorter distances in mitosis, compared to single color two spot measurements.

[Editors' note: further revisions were requested prior to acceptance, as described below.]

*Major point*: *The authors still do not show how their parameters are benchmarked.*

*“Parameters with defined values are benchmarked based on experimentally observed α angles (Luger, 1997; Bednar, 1998; Engelhardt, 2007) and local movements of nucleosomes (Hihara, 2012), as described above*.*”*

*At the minimum, they need to show the experimental observed alpha angle distribution, i.e. probability distribution vs alpha and show that their simulated alpha angle distribution matches the experimental distribution*.

We now show the simulated alpha angle distribution in a new Figure 1—figure supplement 2. We also expanded our description in the subsection headed “Theory and functions of the Type I and II model”:

“the α angle distribution for our simulated chromosomes has a bell shape curve, which peaks at around 70° (benchmarked with experimental measu, based on the nucleosome crystal structure (Luger, 1997) and tails off at around 20° and 180°, consistent with the relative flexibility that is thought to characterize the α angle·(Figure 1—figure supplement 2; Bednar, 1998, Engelhardt, 2007).”

Because it is not currently possible to visualize the path of DNA within chromosomes, no direct experimental measurement of the α·angle distribution is available.

In addition, they need to show, preferentially in figures, how the simulated local movements of nucleosomes match the experimental movements.

The simulated local nucleosome movements are now depicted in a new Figure 1—figure supplement 1 and the following paragraph has been added in the subsection “Theory and functions of the Type I and II model”:

“Nucleosome bead movement is principally regulated by the entropic force and the spring constant of the nucleosome linker […] Key aspects of our model were robust to the exact values of this parameter pair (see 4., below).”

*Minor point: They argue in the rebuttal that 2-5 pN of force should not unravel DNA by referencing a 2001 paper. However, more recent papers in PNAS by Michelle Wang's group and Carlos Bustamante's group showed that DNA indeed unravels at such forces*.

Both Wang’s and Bustamante’s groups observed elastic breathing of nucleosomal DNA, but not unraveling of the nucleosome, at forces of 2-5 pN. This was most recently confirmed in a detailed study by [42]. These authors found that DNA flexibility makes a principal contribution as to whether or not partial elastic unwrapping of the nucleosome occurs at forces in this range. To clarify this, we have added the following sentence in the subsection entitled “Model robustness”:

“We note that the inter-nucleosome tension with either parameter pair is insufficient to disrupt the histone/DNA interaction, though forces in the low pN range may in some cases, depending on the DNA sequence, lead to elastic DNA breathing on the nucleosome surface (Ngo, 2015).”